# Pathogens MenTORing Macrophages and Dendritic Cells: Manipulation of mTOR and Cellular Metabolism to Promote Immune Escape

**DOI:** 10.3390/cells9010161

**Published:** 2020-01-09

**Authors:** Lonneke V. Nouwen, Bart Everts

**Affiliations:** Department of Parasitology, Leiden University Medical Center, 2333 ZA Leiden, The Netherlands; lonnekenouwen@gmail.com

**Keywords:** mTOR, dendritic cells, macrophages, immune escape, cellular metabolism, pathogens

## Abstract

Myeloid cells, including macrophages and dendritic cells, represent an important first line of defense against infections. Upon recognition of pathogens, these cells undergo a metabolic reprogramming that supports their activation and ability to respond to the invading pathogens. An important metabolic regulator of these cells is mammalian target of rapamycin (mTOR). During infection, pathogens use host metabolic pathways to scavenge host nutrients, as well as target metabolic pathways for subversion of the host immune response that together facilitate pathogen survival. Given the pivotal role of mTOR in controlling metabolism and DC and macrophage function, pathogens have evolved strategies to target this pathway to manipulate these cells. This review seeks to discuss the most recent insights into how pathogens target DC and macrophage metabolism to subvert potential deleterious immune responses against them, by focusing on the metabolic pathways that are known to regulate and to be regulated by mTOR signaling including amino acid, lipid and carbohydrate metabolism, and autophagy.

## 1. Introduction

Immune cells from the myeloid lineage, such as dendritic cells (DCs) and macrophages, form an important first line of defense against invading pathogens. DCs have superior antigen presenting capacities and are crucial for the initiation of adaptive immune responses by controlling priming and differentiation of T cell responses. Macrophages on the other hand, are tissue resident cells that excel in clearing of invading pathogens and infected cells (pro-inflammatory or M1 macrophages) and restoring tissue homeostasis (anti-inflammatory or M2 macrophages). Macrophages as well as DCs express a wide range of pattern recognition receptors (PPRs) and other sensors that allow them to directly or indirectly detect the presence of unwanted intruders. This will initiate a cellular activation program that enables them to directly fight the infectious agent, recruit other immune cells, or initiate adaptive immune responses. Recent work has revealed that DC and macrophage activation status is intrinsically linked to metabolic reprogramming [1,2]. In both DC and macrophages, sensing of pathogens, in particular bacteria, is accompanied by a shift toward glycolytic metabolism, the use of the pentose phosphate pathway (PPP), increased translation, as well and altered activity of the TCA cycle. These metabolic changes have been shown to be essential not only for supporting the increased bioenergetic and biosynthetic demand of DC and macrophage activation, but also for generation of metabolites that affect the epigenetic and transcriptional landscapes of these cells that are integral to their functional properties and fate [2,3,4].

Despite the fact that multicellular organisms have developed elegant immunological strategies to combat and counteract infections, pathogens often still manage to successfully infect their hosts. This can in part be attributed to their ability to evade, modulate, and/or suppress these same immune responses [5,6,7]. In this respect, apart from host immune cells using their metabolism to initiate host defense responses, also pathogens are able to exploit these host metabolic pathways for their own benefit. On the one hand, they tap into these pathways to acquire nutrients and macromolecules that directly promote their own survival and replication [8,9,10,11,12,13,14,15]. On the other hand, it is now becoming clear that pathogens also manipulate metabolic pathways of host innate immune cells, to modulate, dampen, or evade the immune responses to further enhance their chances of survival.

A central regulator of cellular metabolism of DC and macrophages is the nutrient sensor mammalian target of rapamycin (mTOR). mTOR consists of two complexes; mTORC1 and mTORC2. mTORC2 regulates cytoskeletal dynamics and activates Akt, a kinase that regulates mTORC1 activity. mTORC1 itself acts as a metabolic rheostat that controls a wide range of cellular processes according to nutrient and energy availability [16]. For example, mTORC1 promotes anabolic processes such as protein translation and represses autophagy under high nutrient availability. Moreover, mTORC1 regulates several metabolic functions of the cell, such as lipid, amino acid, and carbohydrate metabolism in response to a range of stimuli, including growth factors and immune cell activation cues such pathogens and cytokines [17]. This puts mTORC1 as central node in regulation of cellular homeostasis, immune cell function, and innate antimicrobial responses. Given the central role for mTOR in regulating cellular metabolism, it may come as no surprise that pathogens have evolved mechanisms that specifically manipulate mTOR signaling and associated metabolic pathways in order to evade host innate immune responses. Notably, this modulation of mTOR is not only limited to innate immune cells. It is well-known that modulation of mTOR signaling also has major effects on the adaptive immunity such as on the balance between effector versus memory T cells not only in the context of infection, but also autoimmunity and cancer, which is covered by excellent recent reviews [18,19,20]. In the current review we focus on the innate immune response, by seeking to provide an overview and to discuss the most recent insights into how pathogens target DC and macrophage metabolism to manipulate the immune response. Specifically, we will focus on direct modulation of mTOR signaling itself as well as the metabolic pathways that are known to regulate and to be regulated by the mTOR pathway including amino acid, lipid and carbohydrate metabolism, and autophagy.

## 2. Direct Targeting of mTOR Complexes by Pathogens

Pathogens have evolved means to directly modulate mTOR signaling to affect DC and macrophage metabolism and function. An elegant example comes from *Brugia malayi*, an extracellular helminth parasite that causes lymphatic filariasis [21]. Live *B. malayi* microfilariae affect human monocyte-derived DCs by inducing cell death, impairing their ability to make IL-12 and IL-10 and reducing their CD4+ T cell-activating capacity [22]. Consistent with an important role for mTORC1 in supporting IL-12 and IL-10 production, microfilariae of *B. malayi* were found to inhibit mTORC1, by means of secretion of a rapamycin-homolog [22]. Likewise, *Leishmania major*, a parasite that grows in macrophages (in this study the B10R bone marrow-derived macrophage cell line), can cleave mTOR via the protease GP63 which leads to inactivation of mTORC1 and the inhibition of host translation. This inhibition of host translation decreased type I IFN production and inducible nitric oxide synthase (iNOS) expression and consequently enhances the survival of the parasite in a BALB/c mouse model [23].

Instead of inhibiting or degrading mTOR, mTOR can also be relocated to modulate its function. Human cytomegalovirus (HCMV) relocalizes mTOR to the site of viral replication during infection of human foreskin fibroblasts (HFFs)/U373-MG cells (a human glioblastoma-astrocytoma cell line). In the presence of amino acids, mTOR activity correlates with a perinuclear localization, while during amino acid depletion mTOR disperses and loses its activity. HCMV induces perinuclear relocalization of mTOR, mediated through dynein, to maintain mTORC1 activity even in an amino acid scarce environment [24,25]. HCMV is able to infect both macrophages and DCs, but the metabolic and immunological consequences for these cells of HCMV-driven constitutive mTOR activation remain to be determined [26,27].

Also mTORC2 is directly targeted by pathogens. Poxviruses, including vaccinia virus (VacV), were found to target both mTORC1 and mTORC2 in Phorbol 12-myristate 13-acetate (PMA) differentiated Tohoku Hospital Pediatrics-1 (THP-1) cells (a human monocyte cell line) by sequestering raptor (part of mTORC1) and rictor (part of mTORC2) through the structural protein F17, while retaining the phosphorylation of S6K by mTOR. This results in relocation of mTOR to the Golgi where it inhibits cyclic GMP-AMP synthase-stimulator of interferon genes (cGAS-STING) activation and hence the induction of interferon-stimulated genes (IRGs). This enables these viruses to block cytosolic sensing, while retaining mTOR-dependent activation of translation [28,29]. mTORC2 signaling is also targeted by HIV-1 to manipulate macrophage migration. The accessory NEF protein of HIV-1 interacts with POTE ankyrin domain family E (POTEE) that enables NEF to directly bind and activate mTORC2. This activation in turn activates Akt and PKC-α leading to the upregulation of M2 markers and increased migration and invasion of HIV-1 infected PMA differentiated THP-1 cells, that may thereby serve as carriers of HIV-1 across mucosal barriers to help disseminate HIV-1 from primary sites of infection [30]. Overall, both intra- and extracellular pathogens are able to subvert immune responses of DCs and macrophages by directly targeting mTORC1 and/or mTORC2 (Figure 1).

## 3. Targeting of Amino Acid Metabolism by Pathogens

mTOR is a key sensor of amino acids and its activation is dependent on sufficient availability of certain amino acids. Several studies have shown that in innate immune cells, such as macrophages, amino acid availability and their metabolism are intricately linked to cellular activation and polarization in an mTOR signaling-dependent manner [31,32]. It is well-known that amino acid metabolism can be subject to manipulation by pathogens to modulate the immune system for their own survival [33,34]. In this section we focus on the role of a number of key amino acids in innate immune responses against pathogens, how pathogens have devised ways to target amino acid-driven modulation of innate immunity, and how mTOR plays into this (Table 1, Figure 2A).

### 3.1. Tryptophan

Tryptophan is an essential amino acid that is catabolized by three different enzymes: tryptophan 2,3-dioxygenase (TDO), indoleamine 2,3-dioxygenase (IDO), and indoleamine 2,3-dioxygenase-2 (IDO2). IDO, in particular, is associated with infection and induced by IFN type I/II production, prostaglandins as well as microbial components [35,36]. Because pathogens are either tryptophan autotrophs (fully depending on host tryptophan), such as *T. gondii* and the mouse chlamydia strain, or at least benefit from host tryptophan for their development or growth, the host immune response upregulates IDO, decreasing its availability to pathogens [37,38,39]. Additionally, the metabolites arising from tryptophan degradation have antimicrobial effects themselves, thereby aiding in host-defense [40]. Notwithstanding, pathogens have evolved strategies to evade this host defense mechanism. This is illustrated by *M. tuberculosis*, *Francisella tularensis*, and *Chlamydia trachomatis* that are able to synthesize tryptophan de novo or from precursors, to become less vulnerable to this host defense mechanism [9,41].

On the other hand, pathogens might also benefit from IDO expression as long term expression of this enzyme can result in immune suppression. It has been described that the induction of IDO by innate immune cells decreases the pool of effector T cells, while increasing the amount of regulatory T cells [42,43]. In addition, IDO expression results in an anti-inflammatory profile in macrophages and DCs [34,44,45,46]. Given the immune suppressive effect of IDO, some pathogens actively stimulate its expression. HCMV and HPV induce the expression of IDO to dampen pro-inflammatory immune responses. HCMV actively manipulates human monocyte-derived DCs to express IDO by expressing a IL-10 homolog, cmvIL-10, encoded by the gene UL111A [47,48]. Likewise, the HPV16 oncoprotein E7 expressed under control of the keratin 14 promotor in a murine skin graft model induces IDO expression in Langerin negative dermal dendritic cells which mediates immune regulation [49].

Amino acid starvation, in this case tryptophan depletion, leads to activation of a stress response mediated by general control non-repressible 2 (GCN2), a kinase activated by amino acid deprivation that reduces global protein synthesis, and concomitantly to inhibition of the mTOR pathway [50,51,52,53]. This integrated stress response has already been described to regulate T-cell suppression in response to tryptophan depletion and to affect primary murine macrophage and DC cytokine production [34,43,54]. It is worth noting that, apart from tryptophan itself, breakdown products of tryptophan, for example kynurenine and serotonin, are also possible activators of mTORC1 [55,56]. Additionally, kynurenines have been implicated in activating the aryl hydrocarbon receptor (AhR) in T cells which contributes to the generation of regulatory T cells, but also to the capacity of DCs (in this case GM-CSF differentiated bone marrow-derived DCs (GM-DCs)) to generate regulatory T cells [57,58]. Other tryptophan metabolites (e.g., serotonin, kynurenic acid, NAD+, indoles) have been implicated in inflammation as well and could therefore potentially be important during infection [59]. Thus, both the depletion of tryptophan and the generation of its breakdown products can affect mTOR activation and/or contribute to the manipulation of the immune response. While the initiation of the stress response may contribute to the aforementioned anti-inflammatory profiles, GCN2 activation/mTOR inhibition may also be host-protective by activating autophagy which in macrophages can contribute to the killing of intracellular pathogens and to enhance antigen presentation to CD4+ and CD8+ T cells by DCs [53,60,61].

Taken together, tryptophan metabolism has an antimicrobial effect in acute infections, whereas it leads to immunosuppression in chronic infections, in part through the GCN2 and mTOR pathways, which impacts innate immune function. A better evaluation of this balance between immune regulation and host-protection will be required in different infectious settings to fully decipher the role of IDO in host-pathogen interplay.

### 3.2. L-Arginine

L-arginine is a substrate of both nitric oxide synthase (NOS) and arginase [62,63]. The main product of NOS is NO, which is an essential microbicidal metabolite of the innate immune system [64]. The inducible form of NOS (iNOS) is not constitutively expressed, but highly induced upon macrophage activation by bacterial products, such as lipopolysaccharide (LPS) and lipoteichoic acid (LTA), and type I cytokines, including IFNγ. Arginase on the other hand, is primarily induced by type 2 cytokines such as IL-4 and IL-13. Arginase expression metabolizes L-arginine to yield L-ornithine and urea that can be used for the production of polyamines (Figure 3) and thus competes with iNOS for L-arginine [65,66]. The balance in iNOS versus arginase activity and thus the production of NO and polyamines proves to be a target for pathogens to modulate the immune response.

Pathogens modulate this balance to favor arginase over iNOS expression for two main reasons: to reduce NO production and to generate polyamines which can be used for the growth of the pathogen and immune modulation. Multiple pathogens, including *Trypanosoma* sps, *Leishmania* spp, *Toxoplasma gondii*, *Salmonella typhimurium*, *Helicobacter pylori*, *Streptococcus pneumonia*, *Candida albicans*, *M. tuberculosis,* and *Schistosoma mansoni* induce host arginases in mouse macrophages [65,66,67,68,69,70]. Moreover, *Leishmania* spp, *Plasmodium* spp, *C. albicans*, *S. mansoni*, and *H. pylori* (RocF) additionally express their own arginase enzyme to favor polyamine over NO production [65,71,72,73,74,75,76], suggesting that this is a common strategy for pathogens to escape from the deleterious effects of NO. In addition, polyamine synthesis itself has been shown to enhance anti-inflammatory alternative activation (AA) program in macrophages, because of their ability to favor mitochondrial respiration which underpins this type of macrophages polarization [77]. While promotion of such an AA phenotype may prevent direct clearance of pathogens, AA macrophages are important players in tissue repair, thereby serving an important host protective role against tissue damage inflicted by these pathogens.

Not only bacteria and parasites, but also viruses exploit host L-arginine metabolism. Viruses mainly use L-arginine metabolism to enhance translation and increase replication efficiency. Because of the strong positive charge of polyamines, they easily bind to the viral DNA/RNA and are able to accelerate viral protein translation [65,78]. For this reason, multiple viruses manipulate the polyamine pathway by encoding (parts of) the polyamine synthesis pathway and some viruses, at least HCV, induce arginase 1 expression. In the case of HCV, the induction of arginase 1 modulates the immune response: HCV induces arginase 1 expression in CD33+ PBMCs which decreased production of IFNγ by co-cultured human NK cells. This suppression of IFNγ production in the NK cells was due to the inhibition of mTOR through the depletion of L-arginine by the arginase [79]. Interestingly, L-arginine is not only depleted by pathogen-driven host arginase expression, but also by arginine deiminase (ADI). This enzyme is expressed by the parasite *Giardia duodenalis* and L-arginine depletion by this enzyme has been shown to modulate cytokine expression and the phenotype of human monocyte-derived DCs probably via mTORC1 inhibition [80].

L-arginine is one of the amino acids known to stimulate mTOR activity, and its absence reduces mTOR activity and affects innate immune cells, as illustrated by HCV and *G. duodenalis* [81,82,83]. This mTOR inhibition resembles the stress response described for tryptophan. Indeed, L-arginine withdrawal from T-cells results in anti-inflammatory responses via GNC2 activation and mTOR inhibition [34,84,85]. From what has been observed for HCV and *G. duodenalis*, it is possible that this stress response also regulates innate immune cell function. Additionally, mTOR has been shown to increase iNOS translation in RAW264.7, peritoneal macrophages, and GM-DCs [86,87]. Therefore, it is conceivable that mTOR inhibition because of pathogen-driven depletion of L-arginine, leads to a reduced iNOS expression which further limits NO production. In conclusion, arginase expression is induced by a variety of pathogens in order to increase polyamine production and suppress NO production, but murine macrophages and mouse models are overrepresented in these studies and the role of mTOR has not yet been fully eluted.

### 3.3. Asparagine

Certain bacteria express an asparaginase, including *S. typhimurium* and *H. pylori* [88,89,90,91,92]. In the case of *S. typhimurium*, asparaginase expression leads to the inhibition of T cell activation and has been linked to the inhibition of mTOR signaling and autophagy [90]. There are also indications that pathogen-derived asparaginases or modulation of asparagine metabolism may affect macrophage function. For instance, *Erwinia* asparaginase, a bacterial derived oncolytic drug has been described to block pro-inflammatory macrophage responses (inhibition of phagocytosis, pro-inflammatory cytokine production, and MHC-II expression in ANA-1 and RAW264.7 murine macrophages). This effect appears to be dependent on inhibition of autophagy as a consequence of sustained mTOR signaling, which may be related to the stimulatory effects of aspartate on mTOR activity [93,94]. Asparaginases are also known to have glutaminase activity, but to what extent this additional enzymatic activity affects macrophage biology remains to be addressed.

In summary, active manipulation of amino acid metabolism is a strategy widely used by different classes of pathogens to modulate and suppress host innate immune responses. In addition to the amino acids described here, others, such as glutamine and glycine, also play important roles in immune responses [95,96]. Whether pathogens target their metabolism too, to modulate immune responses is still on open question. As illustrated above, active depletion of amino acids by pathogens appears to be a common strategy to compromise innate immune function by reducing mTOR activity. However, it is worth noting that some pathogens such as HCMV and *S. typhimurium*, that rely on host cell mTOR signaling for effective replication, are able to keep mTOR and/or its downstream targets activated even under amino acid-deprived conditions [24,97,98,99,100]. This illustrates that pathogens have developed multiple ways to calibrate host immune cell amino acid metabolism and mTOR signaling that together allow for immune evasion without compromising their own replication.

## 4. Targeting of Lipid Metabolism by Pathogens

mTOR signaling is an important driver of fatty acid (FA) and lipid biosynthesis. Lipids are the major structural component of cellular membranes as well as lipid droplets (LDs). In addition, FAs, particularly when they are polyunsaturated (PUFA), can serve as second messengers and signaling molecules in immune responses. Lipid metabolism in host cells can be modulated and hijacked by pathogens to modify immune responses, which is discussed in this section (Table 1, Figure 2B) [101,102].

### 4.1. Lipid Droplets

LDs are lipid-rich organelles that regulate the storage and hydrolysis of neutral lipids. In the context of immune responses these organelles play several important roles: they help control infection by serving as a docking site for IFN-induced proteins, they are a source of immune modulatory PUFAs such as prostaglandins E_2_ (PGE_2_), and of fatty acids that can be used to fuel Oxphos, and they play a role in cross-presentation [103]. LDs in the host are formed rapidly in response to various infectious agents following PRR signaling [104,105,106]. Given their role in aiding cross-presentation and IFN-signaling, their formation is thought to contribute to the host defense particularly in the context of viral infection [107,108,109].

However, LDs are also important sites of PGE_2_ synthesis, which is known to downregulate pro-inflammatory phenotypes and responses, predominantly in innate immune cells, thereby compromising resistance to multiple infections [106,109,110,111]. For example, *T. cruzi* benefits from the formation of LDs and the subsequent PGE_2_ synthesis since the inhibition of COX-2 by aspirin and NS-398-inhibited LD biogenesis in infected macrophages, suppressed the PGE_2_ production induced by *T. cruzi*, and diminished the enhancement on parasite replication [112]. Similarly, HCMV, HIV, and *M. leprae* infection induce LD formation and the production of PGE_2_ from arachidonic acid. In HIV infection, PGE_2_ has been linked to reduced antigen-stimulated lymphocyte proliferation, while PGE_2_ in HCMV infection promotes successful replication. In *M. leprae* infection, this synthesis is associated with LD formation in Schwann cells, reducing the killing ability of those cells [113,114,115,116,117,118].

In this light, it may come as no surprise that pathogens have been described to actively modulate LD formation and PGE_2_ production. This is exemplified by *S. typhimurium* that stimulates LD formation by secreting the secreted effector protein SseJ and by Chlamydia pneumoniae and *T. cruzi* that are able to accumulate LDs in their vacuoles [104]. Furthermore, the Salmonella pathogenicity island 2 protein C (SpiC) upregulates cyclooxygenase 2 (COX-2) and PGE_2_ synthesis via the ERK1/2 signal transduction pathway in RAW264.7 macrophages, leading to elevated IL-10 expression [60]. COX-2 upregulation and PGE_2_ synthesis were observed to a greater extent when murine bone marrow-derived macrophages (BMDMs) and GM-DCs were infected with viable *S. typhimurium*, both in vitro and in vivo, suggesting that indeed the pathogen itself modulates the production of PGE_2_ [119]. On top of that, human pathogenic fungi also produce PGE_2_ and accumulate LDs themselves which has been associated with increased virulence [120,121,122]. These examples support the idea that pathogens are able to hijack the regulation of LD and PGE_2_ formation to promote immune evasion. However, especially in the case of viruses, in particular HCV and rotaviruses, this might not always be for immunological reasons, but rather to enhance replication efficiency by using LDs as sites of replication and for the assembly of nascent virions [104,123]. Flaviviruses, including Dengue, induce autophagy of LDs, termed lipophagy, to enhance replication efficiency [124]. Dengue induces lipophagy by activating adenosine 5′-monophosphate-activated protein kinase (AMPK), which inhibits mTOR and hence activates autophagy [125].

As already alluded to, mTOR signaling is important for LD formation. The activation of mTOR leads to an upregulation of fatty acid binding proteins (FABPs) that transport free fatty acids (FFAs) to various organelles, and of enzymes involved in triacylglycerol (TAG) synthesis. Consistent with this, *M. tuberculosis* and *T. gondii* have been shown to promote mTORC1-dependent TAG accumulation in human macrophages or other cells that contributes to the LD formation [102,126]. Moreover, mannan and peptidoglycan induce COX-2 expression in human polymorphonuclear leukocytes via PRRs and the activation of the PI3K-Akt-mTOR pathway, implicating mTOR signaling not only in LD formation, but also in COX-2 expression that is essential for the manipulation of the host immune response by pathogens [127].

Interestingly, LD formation is also increased by the inhibition of mTORC1 in an autophagy-dependent manner. Autophagy induction is an alternative for increasing TAG levels by increasing the pool of FFAs of which a portion is immediately re-esterified to form TAGs. These TAGs are subsequently packaged into new LDs [128]. Indeed, during *L. amazonensis* infection of BALB/c macrophages, autophagy in macrophages was associated with increased LD formation, PGE_2_ synthesis, and increased parasite load [129]. Additionally, rapamycin treatment induces LD formation in the yeast *Saccharomyces cerevisiae* [130]. This shows that both activation and inhibition of mTOR signaling can result in the formation of LDs. Whether this results in structurally distinct LDs with different immunological functions requires further study [103,131]. In this context it is interesting to note that *M. leprae* induces LDs that seem to mainly consist of cholesterol and cholesterol esters, while in *M. tuberculosis* infection LD formation is primarily the result of TAG accumulation [132]. It is tempting to speculate that different pathogens promote the formation of LDs via distinct mechanisms to promote LD formation with functions tailored to their needs.

From what has been discussed above, we may conclude that pathogens can often benefit from the formation of LDs and PGE_2_ production. mTOR is involved in the formation of LDs and possibly gives rise to functionally different LDs depending on whether mTOR is activated or not. The different functions and their implications in infection is an area that should be explored in more detail to better understand the role of LDs in pathogen–host interactions.

### 4.2. Membranes

Lipids and fatty acids are an abundant component of membranes and are important for signaling transduction after receptor activation, for example PRRs [133,134]. These receptors are often recruited to and activated in lipid rafts, of which the lipid composition shapes PRR-driven signaling. There are examples of pathogens that are able to modulate lipid composition in either the cytosol or the membranes of target cells in order to alter immune responses. West Nile virus (WNV), for instance, has been shown to redistribute intracellular cholesterol to replication sites, thereby reducing cholesterol levels at the plasma membrane which leads to disruption in antiviral Jak-STAT signaling [135]. Likewise, *L. major* alters macrophage (BALB/c-derived peritoneal macrophages) CD40 signalosome composition by depleting cholesterol, resulting in the induction of IL-10 production rather than IL-12 [136]. This alteration is regulated by TNF receptor associated factor 6 (TRAF6) and spleen tyrosine kinase (SYK) which both have been shown to regulate mTORC1 signaling, suggesting that mTORC1 signaling is involved in the altered CD40 signalosome composition by *L. major* [137,138]. Interestingly, cholesterol activates mTORC1 at the lysosomal surface, which indicates that alterations in cholesterol levels elicited by pathogens could, in addition to membrane composition, alter mTORC1 activation [139]. Since pathogens, predominantly viruses, induce membrane formation for their own replication and/or budding, and membrane formation is stimulated via lipid synthesis by mTOR, it is possible that immune regulation through pathogen-driven modifications of membrane composition to modulate signaling and immune responses are more common than currently known.

## 5. Targeting of Carbohydrate Metabolism by Pathogens

Breakdown of carbohydrates, in particular glucose, plays a central metabolic role in supporting activation and function of immune cells by serving as a rapid source of ATP as well as of carbons to fuel anabolic processes [140,141]. The shift toward anabolism and glycolysis upon macrophage and DC activation is supported by the activation of HIF-1α via mTOR [142]. The involvement of these regulators in glycolysis makes them targets for pathogens aiming to prevent or exploit this shift (Table 1, Figure 2C). For instance, *F. tularensis* prevents the glycolytic shift by downregulating HIF-1α expression and the subsequent upregulation of glycolytic enzyme PFKFB3 (the gene encoding 6-phosphofructo-2-kinase/fructose-2,6-bisphosphatase 3; it catalyzes the synthesis of fructose-2,6-bisphosphate), which is thought to be driven by its capsule that mimics carbohydrates and outcompetes host sugars that are normally used as substrates in aerobic glycolysis. This downregulation of glycolysis ensures optimal replication and results in impaired pro-inflammatory cytokine production in primary macrophages [143]. Pathogens can also interfere with glycolysis by depleting host cells from glucose. *S. typhimurium*, for instance, inhibits glycolysis in murine BMDMs by depleting intracellular glucose [144]. Interestingly, this depletion triggers NLRP3 inflammasome signaling which is potentially detrimental for the pathogen. However, this inflammasome activation can be inhibited by *S. typhimurium* by reducing citrate and reactive oxygen species (ROS) levels [145].

*C. albicans* exploits glucose metabolism by depleting extracellular glucose, instead of intracellular glucose levels. When *C. albicans* exits the macrophage (murine BMDMs and PMA differentiated THP-1 cells) it induces glucose uptake and glycolysis through the transcription activators Tye7 and Gal4, which coincides with macrophage restimulation and upregulation of glycolysis. As a consequence, in the absence of glucose these macrophages fail to upregulate glycolysis which leads to the death of the macrophages, thereby evading immune clearance [146]. *L. infantum* does not prevent the glycolytic shift, but instead promotes glucose oxidation in the mitochondria of murine BMDMs via activation of sirtuin 1 (SIRT1), liver kinase B1 (LKB1), and downstream AMPK, and subsequently via inhibition of mTOR activity. SIRT1 activity is driven by the alteration in intracellular NAD+/NADH ratios. As *L. infantum* is an NAD+ auxotroph, it is likely that depletion of NAD+ underlies the ability of *L. infantum* to activate this signaling route [147]. AMPK activation in macrophages has been shown to prevent the acquisition of an inflammatory profile, which would facilitate survival and proliferation of *L. infantum* [148]. In contrast to the aforementioned pathogens, *L. pneumophila* promotes a Warburg-like metabolism in human monocyte-derived macrophages, which is a consequence of mitochondrial fragmentation and important for bacterial replication [149]. *L. pneumophila* actively sustains mTOR activity (through activation of PI3K by the effector proteins Dot/Icm), which may contribute to the promotion of glycolysis [150,151,152]. These examples illustrate that carbohydrate metabolism, in particular glycolysis, is strongly connected to mTOR signaling and that they are a target for various pathogens to modulate the immune response.

## 6. Targeting of Other mTOR-Controlled Processes by Pathogens

Pathogens do not only target mTOR signaling to alter DC and macrophage metabolism but also to hijack and take advantage of several other processes that are under control of mTOR, such as autophagy and translation, to promote immune evasion. Manipulation of these processes by pathogens are discussed here (Table 1, Figure 2D).

### 6.1. Autophagy

Autophagy is an essential cellular process that is involved in preventing nutritional starvation and in the defense against pathogens [153]. In innate immunity, autophagy is used in several processes including the control of cytokine production, the degradation and killing of bacteria, and the (cross) presentation of foreign peptides by DCs to induce adaptive immune responses [154]. mTOR is a major regulator of autophagy and because of the role of autophagy in innate immunity, pathogens often modulate the mTOR pathway to affect autophagy [155].

Viruses and intracellular bacteria predominantly suppress autophagy directly or by stimulating mTOR activity, to avoid being degraded and/or since they rely on host anabolic metabolism for their replication [156,157,158,159,160,161]. Following infection of murine BMDMs with *S. typhimurium*, for example, targets AMPK, SIRT1, and LKB1 to lysosomes for degradation, thereby preventing inhibition of mTOR and activation of autophagy even though ATP and NAD+ are depleted [98]. *H. pylori* inhibits autophagy as well. Cytotoxin-associated-gene A (CagA) is a critical virulence factor of *H. pylori* that mediates the activation of mTOR and thus the observed inhibition of autophagy [162].

Nonetheless, some pathogens benefit from the induction of autophagy and thus inhibit mTOR rather than activating it. These pathogens include flaviviruses and coxsackievirus that probably use vesicles arising from autophagy as sites of replication, to liberate nutrients and/or to protect host cells from nutrient starvation-induced death [163,164,165,166]. *T. gondii* and *L. donovani* are also inducers of autophagy, but both parasites stimulate autophagy independent of mTOR in order to maintain mTOR activation [167,168]. *L. donovani* even regulates autophagy in a time-dependent manner in PMA differentiated THP-1 cells. In the early stages of infection mTOR is activated and autophagy is decreased, while later in infection autophagy is activated through mTOR-independent mechanisms, possibly to ensure nutrient supply needed to sustain an ongoing infection [169].

Given the important role for autophagy in shaping innate immune cell function, regulation of autophagy by pathogens also has immunological consequences. By stimulating mTOR, HIV-1 can use the inhibition of autophagy to diminish cross-presentation in infected human monocyte-derived DCs, which enables transfer of infection to neighboring cells such as CD4+ T cells from infected DCs [170,171]. In summary, autophagy is both inhibited by pathogens to evade pathogen degradation and immune recognition as well as stimulated to serve as the nutrient source to sustain replication and infection, illustrating that this process is also targeted by pathogens to promote infection.

### 6.2. Host Translation

Various pathogens modulate the induction of translation by mTOR to favor the translation of their mRNAs instead of those of the host, to liberate amino acids or to inhibit cytokine responses [172,173,174,175,176]. For example, *T. gondii* activates mTORC1 to increase translation of mTOR-sensitive transcripts, including those encoding proteins for mitochondrial biogenesis and function. The type I and III strains of *T. gondii* associate with mitochondria during infection and this association impacts cytokine responses (including increased pro-inflammatory IL-6 and CCL5, but also anti-inflammatory IL-10 production) during in vitro infection of BMDMs with *T. gondii* and ex vivo in peritoneal exudate cells (PECs) isolated from infected C57BL/6 mice. Therefore, the mitochondrial association and translation of mitochondrial transcripts via mTOR could be a possible route for the manipulation of the host immune response by *T. gondii* [177,178].

On the other hand, *L. major* cleaves mTOR to inhibit host translation which prevents for the deleterious effects of type I IFN production and iNOS expression [23]. Also, viruses have been shown to inhibit host-translation to suppress anti-viral cytokine production. The vhs protein of herpes simplex virus (HSV)-1 (encoded by the *UL41* gene), for instance, mediates host translation shutoff in the early stage of the virus replication cycle. This shutoff leads to less production of innate cytokines IL-1β, IL-8, and MIP-1 by macrophages [179]. Moreover, mTORC1 inhibition results in the downregulation of the antiviral interferon-induced transmembrane (IFITM) proteins, especially IFITM3. This downregulation of IFITM proteins has already been shown to promote influenza virus A infection in HeLa cells and HFFs [180]. Interestingly, viruses employing internal ribosome entry sites (IRESs) for translation of their mRNAs may benefit from the inhibition of mTORC1, since those viruses do not depend on mTOR-driven cap-dependent translation [23,161,181,182,183].

### 6.3. Innate Cytokines

Apart from autophagy and translation, mTOR regulates cytokine production in innate immune cells; in particular the balance between pro-inflammatory IL-12 and anti-inflammatory IL-10 expression. DCs and macrophages require mTORC1 signaling for IL-10 expression, which suppresses IL-12 production [17,184]. Various pathogens exploit this reciprocal link by elevating IL-10 production. Different cell wall components of *Staphylococcus aureus* can activate toll-like receptor 2 (TLR2) signaling, of which some lead to pro-(TNFα production) and others to anti-inflammatory (IL-10 production) responses by human PBMCs. Specifically, IL-10 production after TLR2 activation is dependent on mTOR signaling illustrating that different moieties of pathogens can be used to induce IL-10 production via mTOR signaling [185]. Likewise, blocking of TLR2 or mTOR attenuated IL-10 production induced by the parasite *L. donovani* in PMA differentiated THP-1 cells while rapamycin decreased IL-10 levels in cornea of *Pseudomonas aeruginosa*-infected BALB/c mice [186,187]. Of note, in spite of the decreased IL-10 levels during *P. aeruginosa* infection, rapamycin treatment increased bacterial loads, suggesting that mTOR dependent processes, other than its role in IL-10 production, are more important in controlling *P. aeruginosa* replication. These examples provide support for the notion that exposure to pathogens can lead to mTOR driven IL-10 expression which could contribute to immune suppression. However, to what extent this mTOR-IL-10 axis is actively exploited by pathogens to promote immune evasion remains to be established.

## 7. Conclusions

In summary, there is a growing body of literature showing that pathogens have evolved elaborate means to directly or indirectly manipulate mTOR signaling to alter DC and macrophage metabolism and function for their own benefit. From these studies it becomes evident that the consequences of stimulating or inhibiting mTOR signaling by pathogens in terms of immune evasion and pathogen persistence and replication are highly context-dependent. This depends on the cell type and the pathogen involved as well as on the stage of infection and at which level mTOR signaling and metabolism is targeted. In addition, it is also a reflection of the fact that mTOR has a myriad of functions ranging from controlling cellular metabolism to regulating protein expression and autophagy.

Despite this complexity, a clear picture is emerging that a vast range of pathogens, from viruses to multicellular parasites, have developed ways to alter mTOR signaling in host DCs and macrophages to shape their metabolism and immune functions to evade immune responses. Nonetheless, many of the mechanisms through which pathogens target and exploit mTOR signaling and host immune cell metabolism for the control of immune cell function remains to be uncovered. Moreover, to what extent certain changes in activity of mTOR signaling and associated metabolic pathways in response to pathogens are a reflection of manipulation by pathogens, a deliberate host response or the sum of the two combined, is often difficult to define and tease apart. Finally, since mTOR also plays a central role in regulating the functional properties of cells of the adaptive immune system, it is likely that some of the immunomodulatory effects induced by pathogens are not only the result of manipulation of mTOR signaling in innate but also adaptive immune cells [188,189,190]. Addressing these issues will not only contribute to the better understanding of the fundamental role of mTOR signaling and metabolism in interplay between host and pathogen, but also has the potential to identify metabolic vulnerabilities of pathogens that can be exploited and targeted in therapeutic settings to treat infections.

## Figures and Tables

**Figure 1 cells-09-00161-f001:**
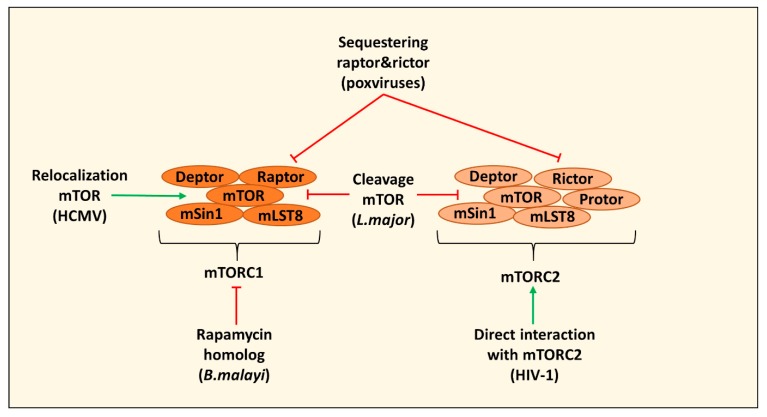
Strategies that pathogens use to directly target mTOR.

**Figure 2 cells-09-00161-f002:**
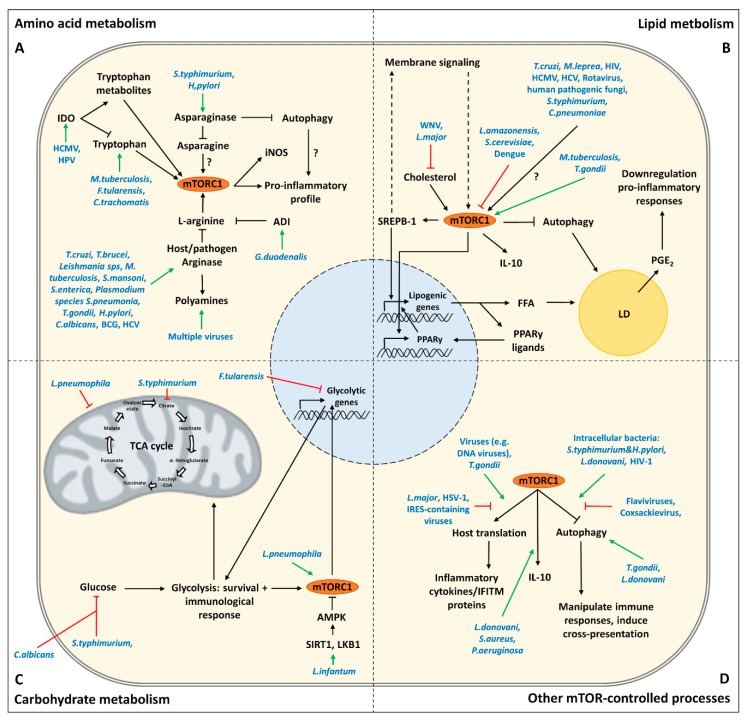
Overview of metabolic targets through which pathogens are known to modulate DC and macrophage functions. In (**A**–**D**), the key strategies through which different pathogens modulate (**A**) amino acid metabolism, (**B**) lipid metabolism, (**C**) carbohydrate metabolism and (**D**) other mTOR-controlled processes in DCs and macrophages to promote immune evasion are indicated Blue = pathogens involved; Red = inhibition by pathogens; Green = stimulation by pathogens..

**Figure 3 cells-09-00161-f003:**
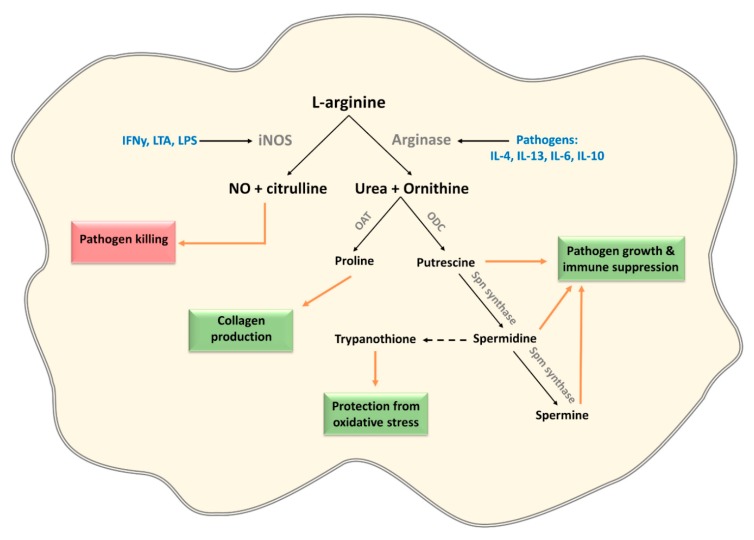
L-arginine is metabolized by iNOS and Arginase, creating competition between pathogen clearance and survival by modulating immune responses through the production of NO and polyamines. The dashed arrow represents a process only observed in trypanosomatid parasites.

**Table 1 cells-09-00161-t001:** Overview of the effects induced by the pathogens described in this review.

Direct Targeting mTOR		
Effect	Pathogen	References
Induction of autophagy by secreting a rapamycin homolog and inhibiting mTOR leading to impaired human monocyte-derived DC function	*B. malayi*	[22]
Cleavage of mTOR via the protease GP63 in B10R macrophages leading to decreased type I IFN production and expression of iNOS	*L. major*	[23]
Relocating mTOR to maintain mTORC1 activation in HFFs and U373-MG cells	HCMV	[24,25]
Sequestering of raptor and rictor in PMA differentiated THP-1 cells leading to mTOR relocalization and inhibition of cGAS-STING activation and induction of IRGs	Poxviruses (including VacV)	[28,29]
Direct interaction with mTORC2 to modulate macrophage phenotype and migration (in this case PMA differentiated THP-1 cells)	HIV-1	[30]
**Amino Acid Metabolism**		
**Effect**	**Pathogen**	**References**
Synthesize tryptophan thereby counteracting IDO depletion of tryptophan	*M. tuberculosis, F. Tularensis, C. trachomatis*	[9,41]
Expresses a IL-10 homolog (cmvIL-10) that induces IDO in human monocyte-derived DCs	HCMV	[47,48]
Induction of IDO in langerin negative dermal dendritic cells	HPV	[49]
Induction of host arginases in mouse macrophages	*Trypanosoma sps*, *Leishmania sps, T. gondii*, *S. typhimurium*, *H. pylori, S. pneumonia, C. albicans, M. tuberculosis* and *S. mansoni*	[65,66,67,68,69,70]
Expression of pathogen arginase	*Leishmania sps, Plasmodium sps, C. albicans, S. mansoni* and *H. pylori*	[65,71,72,73,74,75,76]
Arginase 1 induction in CD33^+^ PBMCs mediates L-arginine depletion leading to mTOR inhibition and decreased IFNy production in co-cultured NK cells	HCV	[79]
Depletion of L-arginine by expression of arginine deiminase modulates cytokine production and phenotype of human monocyte-derived DCs via the inhibition of mTORC1	*G. duodenalis*	[80]
Asparaginase expression leading to asparagine depletion and dampening of immune responses in T cells and macrophages (ANA-1 and RAW264.7 cells)	Expression: multiple pathogens including *H. pylori* and *S. typhimurium* Dampening immune response: *S. typhimurium, Erwinia* asparaginase	[88,89,90,91,92] [90,93]
**Lipid Metabolism**		
**Effect**	**Pathogen**	**References**
Induction of LD formation and PGE2 synthesis for successful replication and modulation of the immune (reduction of antigen-stimulated lymphocyte replication, reduction of killing ability infected cells)	*T. cruzi*, *M. leprae*, HMCV, HIV	[112,113,114,115,116,117,118]
Active stimulation of LD formation via SseJ	*S. Typhimurium*	[104]
Accumulation of LDs in pathogen vacuoles	*C. pneumoniae, T. cruzi*	[104]
Upregulation COX-2 and PGE2 synthesis in RAW264.7 cells and murine BMDMs and BMDCs	*S. typhimurium*	[60,119]
Production of PGE_2_ and accumulation of LDs increasing virulence	Human pathogenic fungi	[120,121,122]
Induction of LDs to enhance replication efficiency and the assembly of nascent virions	Viruses, including HCV and Rotaviruses	[104,123]
mTOR inhibition to induce autophagy of LDs to enhance replication efficiency	Flaviviruses, including Dengue	[124,125]
Promoting mTORC1-dependent TAG accumulation in human macrophages that contributes to LD formation	*M. tuberculosis, T. gondii*	[102,126]
Inducing LDs and PGE2 synthesis via autophagy in BALB/c macrophages	*L. amazonensis*	[129]
Rapamycin induces LD formation	*S. cerevisiae*	[130]
Reducing cholesterol levels at the plasma membrane of Vero cells which disrupts Jak-STAT signaling	WNV	[135]
Altered CD40 signalosome in BALB/c derived peritoneal macrophages by depleting cholesterol leading to IL-10 production	*L. major*	[136]
**Carbohydrate Metabolism**		
**Effect**	**Pathogen**	**References**
Prevents glycolytic shift in primary macrophages by downregulating HIF1α	*F. Tularensis*	[143]
Depleting intracellular glucose to inhibit glycolysis in murine BMDMs	*S. Typhimurium*	[144,145]
Depleting extracellular glucose leading to the death of restimulated macrophages (murine BMDMs and PMA differentiated THP-1 cells)	*C. albicans*	[146]
Promotion of glucose oxidation in murine BMDMs through activation of SIRT1, LKB1 and AMPK increasing survival and proliferation	*L. infantum*	[147]
Promotes glycolysis (Warburg like metabolism) in human monocyte-derived macrophages	*L. pneumophila*	[149,150,151,152]
**Autophagy**		
**Effect**	**Pathogen**	**References**
Inhibition by stimulating the PI3K-Akt-mTOR pathway	Viruses in general	[160,161]
Targeting AMPK, SIRT1 and LKB1 for degradation in murine BMDMs	*S. typhimurium*	[98]
Activation of mTOR (via CagA) and inhibition of autophagy	*H. pylori*	[162]
Induction of autophagy to increase sites of replication, to liberate nutrients and/or to protect host cell death	Flaviviruses (including Zika), coxsackievirus	[163,164,165,166]
Induction of autophagy independent of mTOR which contributes to parasite growth.	*T. gondii*	[167,168]
Inhibition of autophagy by stimulating mTOR early in infection in PMA differentiated THP-1 cells. Induction of autophagy during later stages of infection, regulated independent of mTOR.	*L. donovani*	[169]
Inhibiting autophagy by stimulating mTOR to decrease cross-presentation and enhance spreading of infection.	HIV-1	[170,171]
**Host Translation**		
**Effect**	**Pathogen**	**References**
Activates mTORC1 to promote translation of proteins for mitochondrial biogenesis and function, possibly to modulate the innate immune response (in BMDMs and PECs)	*T. gondii*	[177,178]
Cleavage of mTOR via the protease GP63 in B10R macrophages to prevent IFN type I production and iNOS translation	*L. major*	[23]
Mediates shut off of host translation which leads to a decreased translation of innate cytokines in U937 cells	HSV-1	[179]
Shut off of host translation to favor replication of own genome and to downregulate IFITM proteins	Viruses relying on IRES-dependent translation	[23,161,180,181,182,183]
**Innate Cytokines**		
**Effect**	**Pathogen**	**References**
Cell wall moieties induce the production of IL-10 via TLR2 and mTOR signaling in human PBMCs	*S. aureus*	[185]
TLR2 and mTOR dependent IL-10 production in PMA differentiated THP-1 cells	*L. donovani*	[186]
Rapamycin decreases IL-10 production in cornea of infected Balb/c mice	*P. aeruginosa*	[187]

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
