# Peer review of "Pathogens MenTORing Macrophages and Dendritic Cells: Manipulation of mTOR and Cellular Metabolism to Promote Immune Escape"

_cells, 2020, doi:10.3390/cells9010161_

Round 1

Reviewer 1 Report

This is an interesting review. Figures are clear and well designed.

I have few suggestions:

Please include a paragraph to introduce macrophages and DCs. Also, it might be worth precising that DCs mentioned in this review are monocyte-derived DCs (not "true" DCs). Please revise the sentence lines 69-71 (unclear - incorrect grammar) Please check Figure references so they appear in order in the text. It would be helpful to have a recapitulative table with the following columns: pathogen; cell type; pathway targeted/mechanism; consequence on immune response

Author Response

We would like to thank the reviewer forhis/her constructive feedback and suggestions to improve the quality of our review. Please find a point-by-point response to the comments.

Reviewer 1

This is an interesting review. Figures are clear and well designed.

Specific comments

Please include a paragraph to introduce macrophages and DCs

We thank the reviewer for this suggestion. Some extra information has now been added to the introduction of the review (line 25).

Also, it might be worth precising that DCs mentioned in this review are monocyte-derived DCs (not "true" DCs)

Not all DCs mentioned are monocyte-derived DCs. To increase clarity we specified the type of DCs and macrophages described as much as possible (throughout the text)

Please revise the sentence lines 69-71 (unclear - incorrect grammar)

This sentence has been revised to increase clarity (line 75).

Please check Figure references so they appear in order in the text.

Figure references now appear in order in the text (figure 2 and figure 3 are now reversed).

It would be helpful to have a recapitulative table with the following columns: pathogen; cell type; pathway targeted/mechanism; consequence on immune response

We have included a table (table 1) to summarize the information described in the review (line 476).

Reviewer 2 Report

This is a timely review on the role of mTOR pathway activation in immune espace within the context of metabolic changes during infections. The review is a bit lilted towards the innate immune system, although ample evidence exists for critical roles of mTOR in shaping adaptive immune responses during normal development and inflammmation in the setting of autoimmunity. While tryptophan is mentioned as a metabolic regulator of mTOR activation, very little detail is provided on how this happens. For example, a metabolite of tryptophan, kynurenine, is an activator of mTORC1 (Metabolomics. 11:1157-1174). Moreover, mTOR activation blocks memory CD8 T-cell development that mediate protective immunity against viral infections (Nature. 2009 Jul 2;460(7251):108-12). In turn, mTOR blockade with rapamycin expands memory CD8 T cells and Tregs in patients with lupus (Lancet, 391:1186-1196) and CD8 T cells in patients with lymphoproliferative diseases (J Clin Invest. 2019 Aug 13;130:4451-4463). It would be important to discuss these findings and to address how infections may lead to autoimmunity via mTOR activation and how therapeutic mTOR blockade in autoimmunity and malignancy may alter immunological memory and susceptibility to infections.

Another mechanistic figure should be added on how tryptophan metabolism affects mTOR activation in cells of the innate and adaptive immune system.

Author Response

We would like to thank the reviewer for his/her constructive feedback and suggestions to improve the quality of our review. Please find a point-by-point response to the comments.

While tryptophan is mentioned as a metabolic regulator of mTOR activation, very little detail is provided on how this happens. For example, a metabolite of tryptophan, kynurenine, is an activator of mTORC1 (Metabolomics. 11:1157-1174)

Thank you very much for the suggestion and for pointing out this reference. We have included this reference and additional information in the tryptophan section (line 153) to highlight the role of tryptophan metabolites in regulating mTOR and immunity.

Moreover, mTOR activation blocks memory CD8 T-cell development that mediate protective immunity against viral infections (Nature. 2009 Jul 2;460(7251):108-12). In turn, mTOR blockade with rapamycin expands memory CD8 T cells and Tregs in patients with lupus (Lancet, 391:1186-1196) and CD8 T cells in patients with lymphoproliferative diseases (J Clin Invest. 2019 Aug 13;130:4451-4463). It would be important to discuss these findings and to address how infections may lead to autoimmunity via mTOR activation and how therapeutic mTOR blockade in autoimmunity and malignancy may alter immunological memory and susceptibility to infections.

We appreciate the suggestion, but adaptive immunity is outside the scope of the special issue on innate immune metabolism that our paper is submitted to. However, we agree that the issues raised by the reviewer are of importance for the completeness of our review. Therefore, we highlighted these issues in the introduction (line 62) and the conclusion (line 466).

Another mechanistic figure should be added on how tryptophan metabolism affects mTOR activation in cells of the innate and adaptive immune system.

Although we felt that a separate figure would be redundant, we included the information added to the review concerning the first comment to the summarizing figure (initially figure 3, now figure 2) to clarify the activation of mTOR by tryptophan (line 121).

Round 2

Reviewer 2 Report

None.